# Case of Japanese Marten (*Martes melampus*) Identification by mtDNA Analysis in a Series of Vehicle Cable Damage Incidents

**DOI:** 10.3390/ani15121795

**Published:** 2025-06-18

**Authors:** Reina Ueda, Yuko Kihara, Shin-ichi Hayama, Aki Tanaka

**Affiliations:** Laboratory of Wildlife Medicine, Department of Veterinary Medicine, Nippon Veterinary and Life Science University, 1-7-1 Kyonan-cho, Musashino-shi 180-8602, Japan; m2431@nvlu.ac.jp (R.U.); ykihara@nvlu.ac.jp (Y.K.); hayama@nvlu.ac.jp (S.-i.H.)

**Keywords:** wildlife forensic science, human–wildlife conflicts, mtDNA analysis, species identification, Japanese marten (*Martes melampus*)

## Abstract

In this study, mitochondrial DNA (mtDNA) analysis was performed to identify the animal species involved in an actual case of vehicle component damage that occurred in an urban area of Japan. The results confirmed that such genetic testing can provide scientific evidence demonstrating that damage to man-made objects was caused by animals.

## 1. Introduction

Wildlife forensic science is an academic discipline that contributes to the monitoring of wildlife ecology and associated environmental factors, public concerns, and the resolution of criminal investigations involving wildlife. Investigations in this field employ techniques from morphology, taxonomy, biochemistry, pathology, toxicology, and genetics [1,2]. Among these, DNA analysis plays a particularly crucial role in the examination of wildlife-derived evidence, especially when specimens are processed, degraded, or morphologically unidentifiable [3]. DNA testing is applied to determine the species or individual animal involved in incidents of harm or aggression [4,5], as well as to identify species involved in illegal wildlife trade or processed animal products [6,7]. Notably, a case where cat hair found on clothing was analyzed led to the identification of a suspect in a homicide investigation [4], underscoring the value of DNA analysis as a source of scientific evidence in forensic investigations [8].

Typically, DNA evidence comprises biological materials such as blood, saliva, feces, hair, teeth, and bones, as well as products derived from these tissues [1]. In forensic investigations related to humans, trace DNA has also been recovered from non-biological objects, such as spent bullet casings [1,9]. However, in wildlife forensic science, reports involving the analysis of trace DNA recovered from artificial objects remain scarce.

In wildlife forensic investigations, two major genetic markers are commonly used: mtDNA markers and short tandem repeat (STR) markers. While STR markers are useful for individual identification and parentage testing, mtDNA markers, owing to their high copy number, are particularly effective for analyzing degraded or low-quantity DNA samples. mtDNA markers are also frequently employed in species identification tests [1,6,8].

This study presents a rare case of species identification in a wildlife forensic investigation using mtDNA markers based on trace DNA collected from an artificial object damaged in a property destruction incident.

## 2. Case Description

In February 2025, a series of incidents occurred in a residential area of a city where the speed sensor cables of parked vehicles were repeatedly severed within an area approximately 2 km diameter area. A total of more than 20 cases were confirmed. Based on footage from motion-sensor security cameras installed around the affected area and the observed patterns of cable damage, the possibility of animal interference was initially considered. However, since some of the cuts exhibited sharp, clean edges, the possibility of human involvement could not be ruled out. As a result, a criminal investigation was launched on suspicion of property damage. Subsequently, speed sensor cables were collected from the crime scenes, along with a gauze-like sample, were submitted to our laboratory for analysis. The gauze-like sample, approximately 1 cm in size and composed of thin woven material, had been found near the locations where the incident occurred. Given the circumstances, it was suspected that the material might carry biological traces of animal origin and it was therefore included for forensic examination. At the request of the local police department responsible for the scene, DNA testing was conducted with the aim of identifying the animal species involved in the incidents.

## 3. Materials and Methods

### 3.1. Test Samples

This study utilized a total of 12 samples: ten speed sensor cables that showed signs of damage (sample numbers: 1–4, 6–9, 11, and 12), and two gauze-like samples (sample numbers: 5 and 10). Each sample was assigned an identification number by the police based on the location from which it was collected and individually sealed in separate bags. Figure 1 shows the cable used in this study.

### 3.2. DNA Extraction

All materials were stored in a frozen state from the time of sample receipt until the DNA extraction process. Swab samples were collected by wiping the surface of the damaged areas of the cables with sterile cotton swabs moistened with TAE buffer for 10–30 s. Since multiple sites on each cable sample were suspected to bear animal traces, the swabbing method was employed in this study to efficiently collect DNA samples. For two gauze-like samples, the entire material was directly used for DNA extraction due to the limited sample quantity.

All 12 samples were processed using the QIAamp DNA Investigator Kit (QIAGEN, Hilden, Germany) for DNA extraction. The procedure was conducted in accordance with the manufacturer’s protocol, with the addition of carrier RNA [10], which is known to enhance the recovery of low DNA concentration. The extracted DNA was suspended in 50 μL of Buffer ATE. DNA concentration and purity were measured using a Thermo Scientific NanoDrop Lite spectrophotometer (Thermo Fisher Scientific Inc., Waltham, MA, USA). The prepared DNA solutions were stored at 2–5 °C.

### 3.3. PCR Amplification

Based on surveillance camera footage from the vicinity of the incident site provided by the police, it was deemed probable that the cable damage had been caused by raccoons, domestic cats, or masked palm civets. Therefore, PCR amplification was conducted using species-specific primers targeting these three species: a raccoon (*Procyon lotor*)-specific primer set [11], a primer set designed based on the mitochondrial control region (CR) of the domestic cat (*Felis catus*) [12], and a primer set for the masked palm civet (*Paguma larvata*) that was designed in our laboratory.

The primer pair designed for detecting masked palm civet DNA (hakubisinF, 5′-CCAACATTCGAAAATCTCACCCACTCGCTAAAATT-3′ and hakubisinR, 5′-CCAATGTTTCATGTCTCTGAAAAGGTATATGAACC-3′) amplifies regions corresponding to positions 312–347 bp and 5–39 bp of the complete coding sequence (CDS) of the cytb gene in the mtDNA of *Paguma larvata* (GenBank accession number: AB511054.1).

Additionally, as the area surrounding the incident sites included green spaces where other wild animals such as foxes and weasels may inhabit, further PCR analysis was performed using universal primers known to amplify vertebrate DNA, including such species [13,14], in order to broaden the scope of trace detection. The four types of primers used for PCR in this study are listed in Table 1. The “Size Range (bp)” column indicates the length in base pairs, including the primers themselves.

PCR was conducted using a T100 Thermal Cycler (Bio-Rad, Hercules, CA, USA), selecting primer annealing temperatures optimal for amplification (49–70 °C). The amplification conditions followed the protocol recommended for KOD FX Neo (TOYOBO Co., Ltd., Osaka, Japan).

Copper, the primary material used in cable conductors, releases copper ions, which are known to inhibit PCR reactions. Therefore, this study used KOD polymerase [15], which is considered less sensitive to inhibition by metal ions, to minimize the inhibitory effects potentially caused by copper contamination.

The PCR mixture (25 µL total volume) contained: 1× Buffer for KOD FX Neo, 0.4 mM dNTPs, 0.3 µM of each primer, less than 100 ng/25 µL of template DNA, 1 U/25 µL of KOD FX Neo, and distilled water. PCR cycling conditions were as follows: initial denaturation at 94 °C for 2 min, followed by 30 cycles of denaturation at 98 °C for 10 s, annealing at 49–70 °C for 30 s, and extension at 68 °C for 30 s.

PCR products were subjected to electrophoresis on a 2% agarose gel for 20–25 min (1× TAE buffer, Promega, Madison, WI, USA; Loading Buffer, Nippon Gene, Tokyo, Japan; voltage: 100 V/cm; electrophoresis system: Mupid-2plus, TaKaRa Co., Ltd., Shiga, Japan). After electrophoresis, the gel was stained with ethidium bromide (EtBr, Nippon Gene, Japan) for 15 min, and DNA bands were visualized under UV light to confirm amplification.

### 3.4. DNA Sequencing and Data Analysis

PCR products were purified using the NucleoSpin^®^ Gel and PCR Clean-up kit (Takara Bio, Kusatsu, Japan) according to the manufacturer’s protocol. The purified products were subjected to direct sequencing by Eurofins Genomics Co., Ltd. (Tokyo, Japan). The same primers used for PCR were employed for direct sequencing.

The obtained nucleotide sequences were aligned using MEGA11 (https://www.megasoftware.net/, accessed on 7 May 2024). Primer regions at both ends of the resulting sequences were removed, and the trimmed sequences were then compared against the NCBI database using BLAST (https://blast.ncbi.nlm.nih.gov/doc/blast-news/2025-BLAST-News.html#elasticblast-1-4-0-is-now-available, accessed on 13 June 2025) searches.

## 4. Results

### 4.1. Appearance of the Sample

The cable materials used in this study exhibited various types of damage, including complete severance (Figure 2), incomplete severance with some copper wires still connected (Figure 3), and rupture limited to the outer sheath of the cable (Figure 4). Additionally, the gauze-like sample showed both white and gray areas (Figure 5).

Regarding the cross-sections of the cable samples, most exhibited irregular or jagged edges. However, some samples, such as the one shown in Figure 4, displayed clean and regular cut surfaces.

### 4.2. DNA Analysis

DNA concentration measurements obtained using a NanoDrop Lite spectrophotometer after the extraction process are described in Table 2.

As a result of electrophoresis, no DNA was detected from raccoon (*Procyon lotor*), cat (*Felis catus*), or masked palm civet (*Paguma larvata*). A distinct band was observed following amplification using the primer pair SCPH02500 and SCPL02981. The electrophoresis results are shown in Figure 6. “M” indicates the marker, with reference DNA sizes of 100, 200, and 300 bp from the bottom, increasing in 100 bp increments. “P” indicates the positive control. In this study, DNA extracted from cat (*Felis catus*) tissue, previously confirmed to yield amplification in a prior study [14], was used as the positive control. “N” denotes the negative control. The numbers correspond to individual sample IDs.

Sequencing results were not obtained for four samples: 2, 5, 10, and 12. Faint bands were observed in electrophoresis for samples 2 and 12, suggesting that the amount of genetic material may have been insufficient or that sample degradation may have affected the results. In contrast, no clear amplification was observed for sample 5 and sample 10 during PCR. Both of these were gauze-like samples, and the results differed from those obtained from cable samples directly subjected to damage.

For the remaining eight samples for which sequencing data were obtained, BLAST searches were conducted against the NCBI database. All samples showed a high probability of originating from the Japanese marten (*Martes melampus*). The results are summarized in Table 3. Genetic distances between each sample are described in Table 4.

## 5. Discussion

The materials analyzed in this study were swab samples collected by wiping the damaged surfaces of a speed sensor cable. Among the 12 samples in which mtDNA was successfully amplified using the primer pair SCPH02500 and SCPL02981, eight samples were identified as containing traces derived from the Japanese marten (*Martes melampus*). However, sequence analysis could not be obtained for four samples: samples 2, 5, 10, and 12. For samples 2 and 12, although bands were observed in electrophoresis, they appeared fainter compared to those of successfully sequenced samples. This suggests the possibility that the amount of target DNA originally present at the sampling site was low. Alternatively, the presence of certain enzymes or PCR inhibitors in the samples may have interfered with DNA recovery. Another possibility is that non-specific products were amplified during PCR, resulting in background noise in the sequencing data and ultimately leading to sequencing failure.

This case demonstrates that when animals damage man-made objects, it is possible to identify the responsible species by analyzing genetic material collected from the damaged surface using a swab sample. Although swab sampling, which is widely used in crime scene investigations, is generally considered to yield lower DNA recovery than direct extraction methods [16], this study successfully identified the animal species using DNA analysis from swab samples. One factor contributing to this success may have been the relatively short period of approximately one month between the initial damage and DNA extraction in the laboratory, which likely helped minimize DNA degradation from environmental exposure.

When the origin of a sample is unknown, as in this case, DNA-based species identification is often performed using universal primers targeting the 16S rRNA gene, which enables amplification across a wide range of animal species [17]. However, the primer pair used in this study (SCPH02500 and SCPL02981) has not yet been confirmed to amplify DNA from all animal taxa. Primer specificity and amplification efficiency can vary depending on the target species, and the amount or degradation level of DNA in each sample may further affect results. Therefore, it is important to recognize that the failure to detect DNA does not definitively prove the absence of a particular species, nor does it guarantee the full identification of the predator involved.

Similar cases of damage to man-made objects by martens have been reported in countries such as Luxembourg and Germany, where the stone marten (*Martes foina*) was found to destroy roof insulation of houses [18] or bite rubber and plastic components in car engine compartments [19]. In the canton of Bern, Switzerland, insurance companies reportedly handled 635 damage claims related to martens between 2002 and mid-2006, with annual losses estimated at around EUR 200,000 during that period. Consequently, repeated incidents of marten-related damage to man-made objects have emerged as a significant economic issue.

The marten’s damaging behavior toward car engine compartments may be driven by exploration or play behavior, or possibly by attraction to volatile substances emitted from rubber and plastic components. However, a study suggested that the primary reason martens approach cars was not to seek heat, shelter, or rest, but rather to engage in territorial marking behavior [19]. While numerous studies have been conducted on the stone marten (*Martes foina*), research on the Japanese marten (*Martes melampus*) remains limited. Prior studies have addressed diet, distribution [20], population surveys in peri-urban areas [21], and reports of noise and odor-related disturbances in attics, but no detailed records exist regarding Japanese marten damage to vehicle parts. Consequently, if the damage in this case was indeed caused by a Japanese marten, this study may provide new insight into their interactions with man-made structures.

The Japanese marten (*Martes melampus*) is a small carnivorous mammal of the Mustelidae family endemic to Japan [22]. It includes two subspecies: the Honshu marten (*Martes melampus melampus*), found in Honshu, Shikoku, and Kyushu, and the Tsushima marten (*Martes melampus tsuensis*), native to the Tsushima Islands in Nagasaki Prefecture [23]. A tracking study of the Honshu marten conducted in Mountain Nyukasa, Nagano Prefecture, reported home range sizes from 58.5 to 358.0 hectares between January and August [24]. The approximately 3.14 km^2^ area (2 km in diameter) where damage occurred in this study falls within that range. While variation in range due to individual, sex, habitat, and season exists, the spatial extent of cable damage aligns with known Japanese marten behavior. Furthermore, the fact that most of the cable damage occurred at night may be consistent with the nocturnal habits of the Japanese marten. Considering the shape of the cable cut surfaces, the DNA analysis results, similarities with prior stone marten cases, and the ecological consistency of the Japanese marten, it is reasonable to conclude that the serial speed sensor cable damage in this case was caused by the Japanese marten.

In cases like this, where morphological traits or observations are insufficient to species determination, DNA-based species identification offers a robust scientific approach for analyzing materials [6]. Traditional methods for identifying animals responsible for damaging man-made objects include eyewitness accounts from the incident area [25], monitoring of bait consumption [25], radio tracking [19,26], and trapping [19,25]. Species identification based on DNA amplification from trace evidence at the site is still relatively uncommon. The verification method presented in this study could contribute to future investigations in the field of wildlife forensic science and ecological research, as well as the development of mitigation strategies for human–wildlife conflict.

Accurately identifying the animal species responsible for damage to man-made objects is also important to prevent inappropriate responses driven by misidentification or bias. In this case, without DNA analysis, there was a risk that local feral cats or raccoons, despite having no connection to the damage, could have been erroneously implicated, potentially leading to improper control or culling measures. Scientifically grounded species identification helps avoid such bias and supports appropriate wildlife conservation and management.

The damaged object in this case was a speed sensor cable from a passenger vehicle. These sensors are installed at each wheel and are critical for accurately measuring vehicle speed. Malfunction of this component can reduce driving safety and increase the risk of accidents, making the damage particularly hazardous. Moreover, repeated damage could lead to significant regional economic losses. Therefore, prompt development and implementation of countermeasures is urgently required. While electric fencing has been reported as a short-term countermeasure to exclude martens from buildings in Switzerland [18], such measures are likely impractical in urban environments like those in Japan. More feasible countermeasures may include applying repellents to vulnerable components or installing lights or sound-emitting devices to deter martens, which are known to be wary animals. However, these methods may lose efficacy over time due to habituation, and careful evaluation of toxicity risks to non-target organisms such as humans, pets, and community cats is essential when selecting and applying repellents. Although no single measure can fully prevent damage by Japanese martens, the integration use of multiple strategies may help reduce the frequency of such incidents.

The Japanese marten *(Martes melampus*) is a protected species under the Wildlife Protection and Hunting Law in Japan. In principle, capturing or killing the animal without authorization is prohibited. However, exceptions may be granted by the Minister of the Environment or prefectural governors if damage to ecosystems or agriculture occurs, or if there is academic necessity [27]. Consequently, future countermeasures may thus depend on decisions by relevant authorities.

In this study, through morphological and molecular analysis and comparison with previous marten-related reports, the possibility that the damage was caused by the Japanese marten was highly suggested. Therefore, the methods employed in this study should be recognized as one form of scientific corroboration rather than conclusive evidence. As more cases involving diverse environments and animal species are reported in the future, such approaches could be further applied in wildlife forensic science, helping to improve predictive models of wildlife-induced damage and contributing to the development of effective mitigation strategies.

## 6. Conclusions

The material analyzed in this study comprised swab samples collected from the surface of damaged cables. Using the analytical methods developed in this study, traces of the Japanese marten (*Martes melampus*) were identified in 8 out of 12 samples. Based on a comprehensive evaluation of morphological features, DNA analysis, and ecological characteristics of martens, the series of cable damage incidents examined in this case was found to be consistent with involvement by the Japanese marten.

In wildlife forensic investigations, DNA analysis based on animal-derived trace evidence plays a critical role. Especially when morphological characteristics alone are insufficient for species identification, molecular biological techniques provide an essential and scientifically reliable method for determination. This study represents a new case report contributing to the field of wildlife forensic science.

## Figures and Tables

**Figure 1 animals-15-01795-f001:**
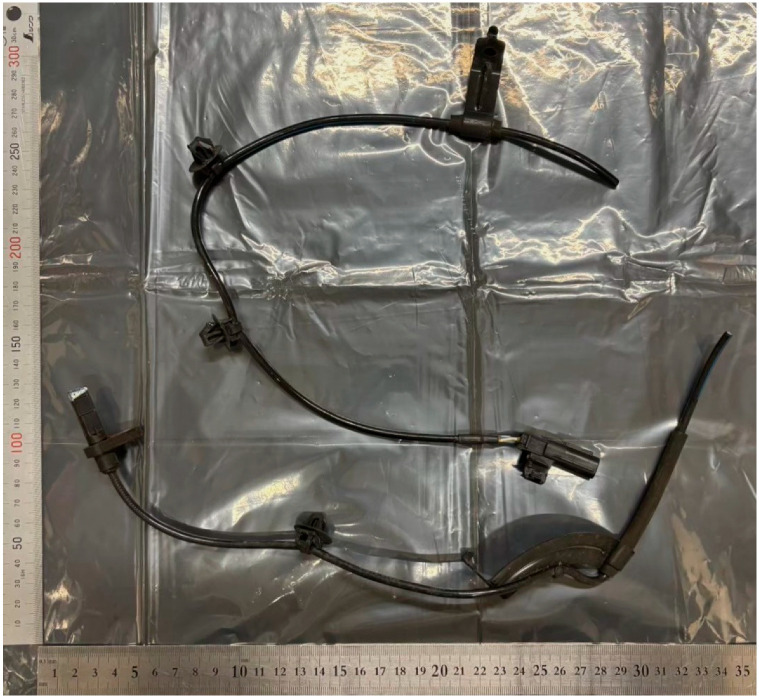
Cable material used in this study.

**Figure 2 animals-15-01795-f002:**
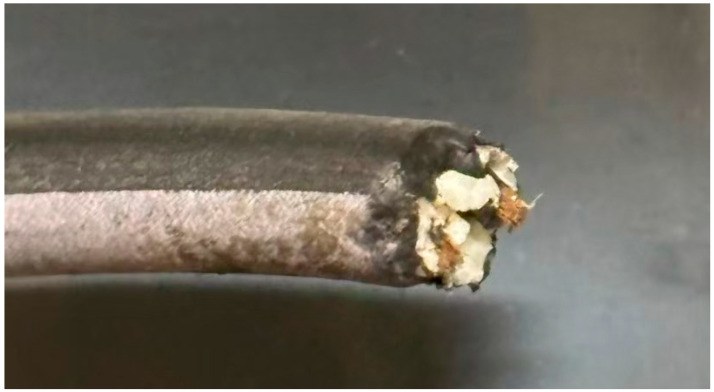
Cable material with complete severance used in this study.

**Figure 3 animals-15-01795-f003:**
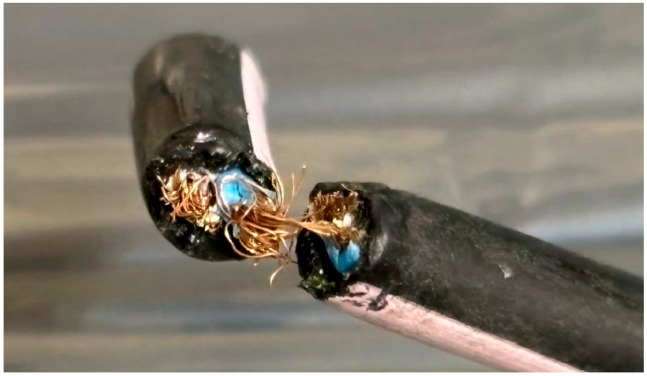
Cable material with incomplete severance used in this study.

**Figure 4 animals-15-01795-f004:**
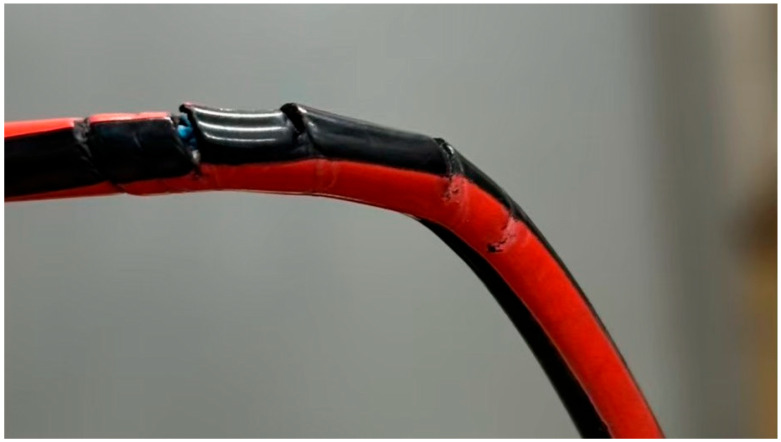
Cable material with sheath rupture used in this study.

**Figure 5 animals-15-01795-f005:**
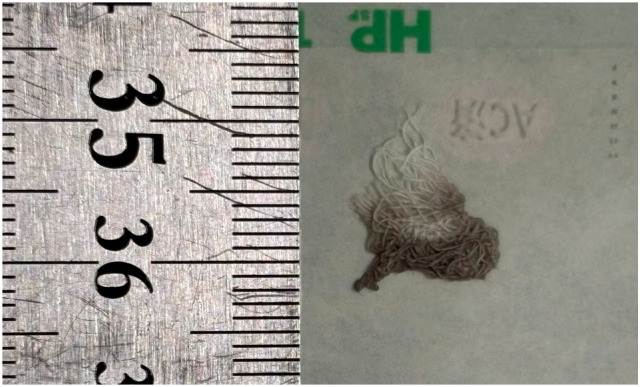
Gauze sample appearance used in this study.

**Figure 6 animals-15-01795-f006:**
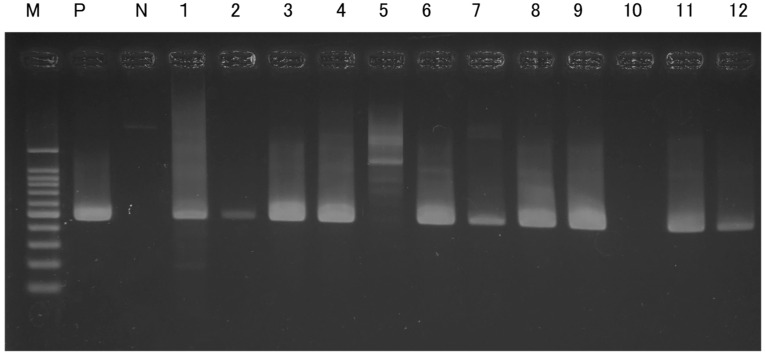
Electrophoresis image of all samples amplified with SCPH02500 and SCPL02981 in this study.

**Table 1 animals-15-01795-t001:** Four primers used for species identification in PCR analysis in this study.

Animal Species	Locus	Forward	Sequence (5′-3′)	Reverse	Sequence (5′-3′)
*Procyon lotor* [11]	mtDNA CR	PLO-L15997	CCATCAGCACCCAAAGCT	PLO-CRL1	CGCTTAAACTTATGTCCTGTAACC
*Paguma larvata*	mtDNA cytb	hakubisinF	CCAACATTCGAAAATCTCACCCACTCGCTAAAATT	hakubisinR	CCAATGTTTCATGTCTCTGAAAAGGTATATGAACC
*Felis catus* [12]	mtDNA CR	JHmtF3	GATAGTGCTTAATCGTGC	JHmtR3	GTCCTGTGGAACAATAGG
Mammals and amphibians [13,14]	16S rRNA	SCPH02500	TTACCAAAACATCACCTCT	SCPL02981	ATCCAACATCGAGGTCGTAA

**Table 2 animals-15-01795-t002:** DNA concentration measurements obtained using a NanoDrop Lite spectrophotometer in this study.

Sample Number	DNA Concentration (ng/µL)
1	14.9
2	10.5
3	17.4
4	14.3
5	15.4
6	11.5
7	11.6
8	14.6
9	14.4
10	16.8
11	11.2
12	14.5

**Table 3 animals-15-01795-t003:** Query sequence lengths, closest matching species, percent identity, and accession numbers for all eight samples used in this study.

Sample Number	Query Length (bp)	Animal Species	Per.ident (%)	Accession
1	463	*Martes melampus*	99.78	AB291076.1
3	463	*Martes melampus*	99.78	AB291076.1
4	463	*Martes melampus*	99.78	AB291076.1
6	467	*Martes melampus*	99.72	AB291076.1
7	463	*Martes melampus*	99.78	AB291076.1
8	463	*Martes melampus*	98.49	AB291076.1
9	463	*Martes melampus*	99.57	AB291076.1
11	463	*Martes melampus*	98.70	AB291076.1

**Table 4 animals-15-01795-t004:** Genetic distances between each sample calculated by “Compute Pairwise Distances” function in MEGA11.

	Reference	1	3	4	6	7	8	9
Reference								
1	0.0022							
3	0.0022	0						
4	0.0022	0	0					
6	0.0065	0.0043	0.0043	0.0043				
7	0.0022	0	0	0	0.0043			
8	0.015	0.013	0.013	0.013	0.018	0.013		
9	0.0043	0.0022	0.00212	0.0022	0.0065	0.0022	0.015	
11	0.0044	0.0022	0.0022	0.0022	0.0066	0.0022	0.015	0.0044

## Data Availability

The original contributions found in this study are included in this article. Further inquiries can be directed at the corresponding author.

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
