# Peer review of "Case of Japanese Marten (*Martes melampus*) Identification by mtDNA Analysis in a Series of Vehicle Cable Damage Incidents"

_animals, 2025, doi:10.3390/ani15121795_

Round 1
Reviewer 1 Report
Comments and Suggestions for Authors
I have several comments on the submitted manuscript:
1) The introduction does not mention plant identification, even if this is integral to wildlife forensics.
2) Only 1 review references Species identification tests. Primary citations are a must for even undergraduate students.
3) The methodology (sample selection) does not consider intact cables (those that have not been attacked by animals).
4) Blind wiping of a surface is an obsolete approach to sample collection. Latent stain visualization (e.g., using DiamondDye) would be appropriate to eliminate the sampling bias (negative results).
5) I miss the quantitation step. An appropriate test would detect the presence of inhibitors and/or degradation and the amount of amplifiable animal DNA.
6) Using species-specific primers is understandable, but why without the main "Martes Sp. suspect"?
7) Relying only on one mtDNA target (16S) and one (NCBI) database is questionable. There is too much space for a bias. See your discussion on l.202.
Author Response
Thank you very much for your comments to help us improve our manuscript. We address each of your comments below and the line number of the revised manuscript is shown:
Comments 1: The introduction does not mention plant identification, even if this is integral to wildlife forensics.
Response 1: Thank you for pointing this out. This study is a forensic veterinary investigation focusing on wild animals and does not involve plants. As plants fall outside our area of expertise, we have never included them in our research. We would greatly appreciate it if this study could be understood strictly as a veterinary investigation.
Comments 2: Only 1 review references Species identification tests. Primary citations are a must for even undergraduate students.
Response 2: Thank you for your comments. We have revised the references so that there are now three citations related to the animal species identification test in Line 51-52 in Introduction section.
Comments 3: The methodology (sample selection) does not consider intact cables (those that have not been attacked by animals).
Response 3: Thank you for your comments. This study is a case report in the field of wildlife forensic veterinary medicine, involving multiple incidents in which cable damage was suspected to have been caused intentionally by humans—potentially constituting property damage offenses. Whether or not undamaged cables showed traces of wild animals was not a subject of investigation, as it falls outside the scope and objectives of this study. Therefore, no such examinations were conducted.
Comments 4: Blind wiping of a surface is an obsolete approach to sample collection. Latent stain visualization (e.g., using DiamondDye) would be appropriate to eliminate the sampling bias (negative results)
Response 4: Thank you for your comments. Using fluorescent dyes such as DiamondDye to preliminarily identify the localization of DNA is an efficient approach and may be appropriate in cases where the samples cover a wide area and potential DNA deposition sites cannot be readily predicted. However, in the present study, the samples collected in our laboratory were intended specifically for determining the cause of cable severance. Therefore, DNA testing was limited to the area immediately surrounding the cut site. Given this narrow scope, we considered the swab method to be an appropriate and sufficient sampling technique.
Comments 5: I miss the quantitation step. An appropriate test would detect the presence of inhibitors and/or degradation and the amount of amplifiable animal DNA.
Response 5: Thank you for your valuable comments. We have added the DNA concentration measurements obtained using a NanoDrop Lite spectrophotometer after the extraction process to Table 1 in Line 171-184 in Result section. While publications on animal forensic genetics describe quantification of isolated DNA as a standard step in species identification workflows, they do not specifically mention the need to test for the presence of inhibitors or degradation products. Accordingly, such tests were not performed in this study (Linacre, A. Animal Forensic Genetics. Genes (Basel) 2021, 12 (4). DOI: 10.3390/genes12040515).
Comments 6: Using species-specific primers is understandable, but why without the main "Martes Sp. suspect"?
Response 6: Thank you for your comments. In our laboratory, we conduct necropsies and DNA analyses upon official request, primarily to assess the possibility of illegal killings involving domestic wildlife. Routine DNA-based species identification is carried out using species-specific primers for animals such as raccoons, crows, masked palm civets, and cats. The range of target species is limited to those with wide distributions and large population sizes in Japan, due to the financial constraint that all testing costs must be covered by our research budget, making it impractical to include a broader range of species. As noted, in this study we were unable to conduct species-specific PCR for the Japanese marten (Martes melampus) due to sample limitations. However, this study is presented as a case report in which a non-routine species was successfully identified using a non-species-specific method. We believe that this case has value as an applied example in wildlife forensic veterinary medicine, particularly under budget-constrained conditions where routine tests do not cover all potentially relevant species.
Comments 7: Relying only on one mtDNA target (16S) and one (NCBI) database is questionable. There is too much space for a bias. See your discussion on l.202.
Response 7: Thank you for pointing this out. This study is a case report involving multiple incidents in which cable damage was suspected to have been caused intentionally by humans, potentially constituting property damage offenses. Although surveillance footage made human involvement appear unlikely, there was no conclusive evidence indicating the involvement of wild animals. As a result, the police requested DNA testing. Accordingly, the aim of this case was not to identify all wildlife species potentially responsible for the cable damage, but rather to determine the presence or absence of wildlife traces and to present this information as scientific evidence within a limited time frame and budget. We believe this approach aligns with the objectives of wildlife forensic veterinary medicine, which seeks to maintain public order without causing unnecessary concern or confusion in society.
Reviewer 2 Report
Comments and Suggestions for Authors
The manuscript titled "Case of Japanese marten (Martes melampus) identification by mtDNA analysis in a series of vehicle cable damage incidents" reports a case attributed to Martes melampus. Understanding this case will likely be of significant interest to readers. The paper is well-written. I recommend that the authors include additional information on genetic distance and sequence differentiation in the manuscript. Furthermore, it would be valuable if the authors could deduce the number of individual Japanese martens involved based on the available data. All references for the primers should be included in Table 1. Ensure that all references pertaining to the software are included.
Some minor revisions have been noted in the PDF document. Please refer to them for further improvements.

Author Response
Thank you very much for your comments to help us improve our manuscript. We address each of your comments below and the line number of the revised manuscript is shown:
We sincerely appreciate your comments on the English usage in the manuscript. We have carefully reviewed and corrected all of the expressions you pointed out.
In addition, all comments indicated in the manuscript have been thoroughly addressed and the necessary revisions have been incorporated.
Comments 1:. Table, all references of primers should be added in the table 1.
Response 1: Thank you for your comments. We have revised the reference, however, the species-specific primer for the masked palm civet (Paguma larvata) was originally designed by our laboratory (Line 104-109), and as such, there is no published reference available.
Comments 2: Table 2, It is better to add the genetic distance of all sequences. If there is existed difference genetic distance or different base, the authors can deduce the number of Japanese marten.
Response 2: We created a table showing the genetic distances between each sample and added it as Table 4. The table was generated using the “Compute Pairwise Distances” function in MEGA11.
Comments 3: 5. Discussion, add the reason why some sample are failed to obtain the DNA sequence.
Response 3: Thank you for pointing this out. We have added “However, sequence analysis could not be obtained for four samples: Samples 2, 5, 10, and 12. For Samples 2 and 12, although bands were observed in electrophoresis, they appeared fainter compared to those of successfully sequenced samples. This suggests the possibility that the amount of target DNA originally present at the sampling site was low. Alternatively, the presence of certain enzymes or PCR inhibitors in the samples may have interfered with DNA recovery. Another possibility is that non-specific products were amplified during PCR, resulting in background noise in the sequencing data and ultimately leading to sequencing failure.” in the Discussion section, Line 225-232.